# TIPS Modification in the Management of Shunt-Induced Hepatic Encephalopathy: Analysis of Predictive Factors and Outcome with Shunt Modification

**DOI:** 10.3390/jcm9020567

**Published:** 2020-02-19

**Authors:** Philipp Schindler, Leon Seifert, Max Masthoff, Arne Riegel, Michael Köhler, Christian Wilms, Hartmut H. Schmidt, Hauke Heinzow, Moritz Wildgruber

**Affiliations:** 1Institute of Clinical Radiology, University Hospital Muenster, D-48149 Muenster, Germany; Philipp.Schindler@ukmuenster.de (P.S.); Max.Masthoff@ukmuenster.de (M.M.); arne.riegel@ukmuenster.de (A.R.); michael.koehler@ukmuenster.de (M.K.); 2Department of Gastroenterology and Hepatology, University Hospital Muenster, D-48149 Muenster, Germany; LeonLouis.Seifert@ukmuenster.de (L.S.); christian.wilms@ukmuenster.de (C.W.); hepar@ukmuenster.de (H.H.S.); hauke.heinzow@ukmuenster.de (H.H.); 3Department of Radiology, University Hospital Ludwig-Maximilians-Universität—Campus Großhadern, D-81377 Munich, Germany

**Keywords:** hepatic encephalopathy, portasystemic shunt, transjugular intrahepatic, liver cirrhosis

## Abstract

Purpose: To evaluate predictive parameters for the development of Hepatic Encephalopathy (HE) after Transjugular Intrahepatic Portosystemic Shunt (TIPS) placement and for success of shunt modification in the management of shunt-induced HE. Methods: A retrospective analysis of all patients with TIPS (*n* = 344) has been performed since 2011 in our university liver center. *n* = 45 patients with HE after TIPS were compared to *n* = 48 patients without HE after TIPS (case-control-matching). Of *n* = 45 patients with TIPS-induced HE, *n* = 20 patients received a reduction stent (*n* = 18) or TIPS occlusion (*n* = 2) and were differentiated into responders (improvement by at least one HE grade according to the West Haven classification) and non-responders (no improvement). Results: Older patient age, increased serum creatinine and elevated International Normalized Ratio (INR) immediately after TIPS placement were independent predictors for the development of HE. In 11/20 patients (responders, 55%) undergoing shunt modification, the HE grade was improved compared with nine non-responders (45%), with no relevant recurrence of refractory ascites or variceal bleeding. A high HE grade after TIPS insertion was the only positive predictor of treatment response (*p* = 0.019). A total of 10/11 responders (91%) survived the 6 months follow-up after modification but only 6/9 non-responders (67%) survived. Discussion: Older patient age as well as an increased serum creatinine and INR after TIPS are potential predictors for the development of HE. TIPS reduction for the treatment of TIPS-induced HE is safe, with particular benefit for patients with pronounced HE.

## 1. Introduction

Transjugular intrahepatic portosystemic shunt (TIPS) insertion is an established method in the management of decompensated liver cirrhosis [1,2]. With technical progress and increasing experience, peri-interventional complications and TIPS-associated mortality were substantially reduced [1,3]. However, hepatic encephalopathy (HE) is one of the most common complications with an incidence of 20% to 50% [4,5,6]. It is based on a multifactorial mechanism of reduced hepatic filter function in liver dysfunction and splanchnic blood shunting into the systemic circulation, as well as an overproduction of intestinal neurotoxins and an increased permeability of the blood–brain barrier [7,8]. Clinically, HE ranges from mild confusion to coma and is commonly graded by the West Haven classification [9,10].

Historically, patients with post-TIPS HE were treated with a low-protein diet as well as nonabsorbable antibiotics and disaccharides to reduce intestinal neurotoxin production and absorption, which is increasingly being questioned today [11,12]. However, up to 8% of patients develop refractory HE after TIPS and there is only limited evidence about specific predictive risk factors for this purpose [13,14,15,16]. In addition, refractory HE after TIPS is often associated with further deterioration of liver function and remains a clinical challenge [17]. In these cases, TIPS modification is the only therapeutic option besides liver transplantation [13,18,19]. TIPS modification reduces the portosystemic shunt volume and can improve HE [20]. Nonetheless, this potential benefit must be critically contrasted with the increased risk of recurrence of the primary TIPS indication (variceal bleeding, refractory ascites) [20]. There are only limited data available regarding the safety, effectiveness and outcome of TIPS modification [14,18,19].

The aim of our study was to evaluate predictive parameters for the development of HE after TIPS placement. Additionally, we aimed to determine the technical and clinical success of TIPS modification in the treatment of shunt-induced HE.

## 2. Materials and Methods

### 2.1. Study Design

The study protocol has been approved by the local ethics committee and all patients provided written informed consent (protocol number 2016-046-f-S, approved by the ethics committee of the Westfälische Wilhelms Universität Münster, Germany).

In this retrospective study, the clinical data of all patients who underwent TIPS in cases of decompensated liver cirrhosis since 2011 in our university liver center (*n* = 344) were collected. *N* = 45 patients (case group) with HE after TIPS were identified and included in the study. To evaluate predictive parameters for the development of HE after TIPS, *n* = 48 consecutive patients (control group) without HE after TIPS were identified by case-control-matching (propensity score matching) regarding age, sex, etiology of liver disease, child grade and indication for TIPS insertion. Laboratory data (Bilirubin, Albumin, INR, Creatinine) were obtained and MELD-, as well as CLIF-c AD-scores [21], were calculated for the day before and within one week after TIPS placement. Invasive portosystemic pressure gradient measurements were obtained within the procedure before and after TIPS placement. Patients’ follow-up was obtained at 1 month, 3 months, 6 months and 1 year after TIPS insertion, and consecutive follow-up was performed annually. Diagnosis of HE was based on the West Haven classification, as follows [9,10]:Grade 0 (minimal)—normal state of consciousness, objectifiable only by neuropsychiatric tests;Grade 1—slight mental slowdown, disturbed fine motor skills;Grade 2—increased fatigue, apathy, flapping tremor/asterixis, ataxia, slurred speech;Grade 3—somnolence, marked disorientation, rigor, stupor;Grade 4—coma.

For assessment of hepatic encephalopathy, patients routinely underwent psychometric testing (number connection test).

Demographic and laboratory data, as well as invasive pressure measurements, were used in logistic regression analysis for independent predictors of post-TIPS HE.

We next studied potential predictors for the response to shunt modification. Of *n* = 45 patients with TIPS-induced HE, we identified *n* = 20 patients with refractory HE who received a TIPS modification (*n* = 18 with reduction stent or *n* = 2 with TIPS occlusion). All patients who received shunt modification had undergone maximal medical treatment previously, including lactulose and rifaximin (Rifaximin 550 mg 1-0-1, Lactulose 3× 15–20 mL), according to current guidelines [22]. Patients were assessed for the development of post-TIPS HE over 4 weeks and divided into responders (improvement by at least one HE grade according to the West Haven classification, *n* = 11) and non-responders (no improvement, *n* = 9). Again, laboratory data (Bilirubin, Albumin, INR, Creatinine) were obtained and MELD-, as well as CLIF-c AD-scores, were calculated the day before and within two to four weeks following TIPS modification. Invasive portosystemic pressure gradient measurements and fluoroscopy time were obtained in the TIPS modification procedure. Recurrence of primary TIPS indication (variceal bleeding, refractory ascites) was recorded. To evaluate potential predictors of response to shunt modification, HE grade, laboratory data, and invasive pressure measurements before and after modification procedure were used for subsequent logistic regression analysis.

### 2.2. TIPS Placement and Modification Procedure

TIPS placement was performed according to standard operating procedures under general anesthesia. Single-shot antibiosis was performed by administration of 2 g ceftriaxone intravenously. All TIPS were done using PTFE-covered Stents (Viatorr, Gore Inc. Flagstaf, AZ, USA). Initially, shunts were set at an 8 mm diameter; in cases of insufficient reduction of the portosystemic gradient (> 12 mmHg) the shunt lumen was increased to 10 mm. Shunt modifications were performed under analgosedation using reduction stents with a 5 mm lumen diameter (Sinus, Optimed, Ettlingen, Germany). Complete shunt occlusions were exerted using Amplatzer Plug Typ II (St. Jude Medical, Saint Paul, MN, USA).

### 2.3. Definitions regarding HE

Refractory HE: persisting HE, which did not clinically improve despite maximal medical management before shunt modification, described elsewhere [18].

Early post-TIPS HE: HE development within 6 months after TIPS placement.

Late post-TIPS HE: HE development more than 6 months after TIPS placement.

### 2.4. Statistics

All data are presented as the mean (SD), median (range) or percentage. The Student *t*-test was used for quantitative data. The Mann–Whitney U-test was used for non-normally distributed, qualitative data. The chi-square test was used for contingency tables. Case-control-matching was performed by propensity score-matching method with SPSS fuzzy extension. Propensity score-matching was performed via binary logistic regression to create a propensity score for each patient, entering the following variables: age, sex, etiology of liver disease, child grade and indication for TIPS insertion. Subsequently, a case-control match between the ‘HE’- and ‘no HE’-group was obtained by use of the nearest-neighborhood-matching using a caliper width of 0.2 without replacement [23]. We used multiple caliper widths from 0.1 to 0.8 of the pooled standard deviation of the logit of the propensity score, in increments of 0.1, to evaluate the proper caliper, which meant the maximum distance that two units could be apart from each other was 0.8, 0.7, 0.6, 0.5, 0.4, 0.3, 0.2, or 0.1 of standard deviations of the logit of the estimated propensity score. Finally, we identified the caliper of 0.2 affording a superior performance in the estimation of both preferred homogeneity and minor loss of sample size. Logistic regression analysis was used to include demographic, clinical and periprocedural data, identifying potential predictors for the development of HE and response to shunt modification. Two-sided *p*-values < 0.05 were considered to be statistically significant. Statistical analysis was performed using the SPSS Statistics version 26 (SPSS Inc., Chicago, IL, USA).

## 3. Results

### 3.1. Patient Characteristics

A total of 344 consecutive patients who underwent TIPS in cases of decompensated liver cirrhosis since 2011 in our university liver center were analyzed retrospectively. Forty-five patients with new or deteriorated HE after TIPS were included (case group, median age, y: 56 (25–77); *n* = 32 male, *n* = 13 female; Table 1). A total of *n* = 48 patients without HE after TIPS were identified for the matched control group (median age, y: 54 (21–77); *n* = 33 male, *n* = 15 female). In both groups, the most frequent etiology of liver disease was alcohol abuse, while liver disease was mostly frequently graded as child B. Refractory ascites was the most frequent indication of TIPS insertion. As Table 1 reveals, there were no significant differences regarding sex, median age, etiology of liver disease, child’s grade and indication for TIPS insertion between the “HE” and the “control” group.

### 3.2. Laboratory and Procedural Data Regarding TIPS Insertion

Liver function parameters were recorded before and after TIPS placement. As shown in Table 2, there were no significant differences regarding pre- and post-procedural Bilirubin, Albumin, Creatinine, International Normalized Ratio (INR) and MELD- or CLIF-c AD-score between both groups. Moreover, there was no statistical significance in the course of liver function regarding TIPS insertion within each group. In all cases, TIPS placement was performed successfully and mean portosystemic gradient (PPG) reduction was 9.9 (± 4.8) mmHg in the “HE” group vs. 9.4 (± 4.5) mmHg in the “control” group, again with no statistical difference (*p* = 0.654).

### 3.3. Logistic Regression Analysis of Predictors for Post-TIPS HE

To evaluate predictive clinical and technical factors for the development of HE, logistic regression analysis of the patients’ demographic, laboratory and procedural data was performed. We identified age (*p* = 0.007), post-TIPS Creatinine (*p* = 0.019) and post-TIPS INR (*p* = 0.040) as independent predictors for the development of post-TIPS HE, as shown in Table 3.

### 3.4. Shunt Modification in Patients with HE

Of 45 patients with TIPS-induced HE, *n* = 20 patients with medically refractory HE underwent shunt modification (*n* = 18 shunt reduction, *n* = two shunt occlusion). None of them had HE before TIPS insertion. Patients who later underwent shunt modification showed a better liver function relative to the patients (*n* = 25) with post-TIPS HE who did not underwent shunt modification (Table 4). Albumin showed significant differences between both groups before TIPS insertion (3.2 vs. 2.2 g/dL, *p* = 0.014). After TIPS placement, there were no statistical differences between both subgroups regarding Albumin levels as well the other laboratory values. Moreover, mean PPG was significantly lower in post-TIPS HE patients who later each underwent shunt modification before (*p* = 0.016) and after (0.027) TIPS placement.

### 3.5. Comparison of Responders and Non-Responders to TIPS Modification

The *n* = 20 patients undergoing shunt modification (*n* = 18 shunt reduction, *n* = two shunt occlusion) were differentiated into *n* = 11 responders (improvement by at least one HE grade according to the West Haven classification; median age, years: 64 (56–77); *n* = nine male, *n* = two female) and *n* = nine non-responders (no improvement; median age, years: 65 (26–76); *n* = three male, *n* = six female). Table 5 reveals demographic, laboratory and procedural data of both groups. *N* = 10/11 patients (90.9%) of the responder group had severe post-TIPS HE before shunt modification (West Haven grade 3 or 4), while *n* = 9/9 patients (100%) of the non-responder group had only moderate post-TIPS HE (West Haven grade 1 or 2). Moreover, the majority of the later responders developed early HE within 6 months after TIPS placement (*n* = 8/11, 72.7%), while the majority of the non-responders developed late post-TIPS HE (*n* = 5/9, 55.6%).

There were no significant differences regarding pre- and post-procedural Bilirubin, Albumin, Creatinine, INR and MELD- or CLIF-c AD-score between responders to TIPS modification and non-responders. Moreover, there was no statistical significance in the course of liver function regarding TIPS modification within each group. In all cases, TIPS modification was performed successfully, with PPG increasing from 7 (1–17) to 12 (6–21) mmHg. The mean PPG increase was 3.8 (± 4.3) mmHg in the responder group vs. 5.6 (± 4.9) mmHg in the non-responder group (*p* = 0.654). No further shunt reduction was performed in the non-responders. The recurrence of refractory ascites or variceal bleeding after shunt modification as the primary TIPS indication was only observed in one case of the responders and two cases of the non-responders (*n* = 1 shunt reduction, *n* = 2 shunt occlusion). Ten responders (91%) survived the 6 months follow-up after modification compared to only six non-responders (67%), *p* = 0.189.

### 3.6. Analysis of Predictors for Response to Shunt Modification

Again, logistic regression analysis of the patients’ demographic, laboratory and procedural data was performed, now regarding shunt modification. Here, only the HE grade after initial TIPS placement revealed to be a positive predictor of treatment response (*p* = 0.019, Table 6), regardless of liver function and procedural data.

## 4. Discussion

TIPS is an established treatment option in decompensated liver cirrhosis to control variceal bleeding or ascites, while HE is one of the most common concerns among TIPS patients, significantly limiting its effectiveness [1]. With an incidence of up to 50%, post-TIPS HE continues to be a clinical challenge, as pharmacological prophylaxis has not shown a significant benefit in preventing HE after TIPS [5,6,24,25]. Therefore, it is of great clinical importance to identify predictive factors for the development of HE after TIPS, which allows patient selection based on a specific risk profile as well as targeting potential risk factors right after TIPS insertion.

This study shows that older patient age, as well as elevated serum creatinine levels and INR in the first week after TIPS, are predictors for the development of HE in the future course. This is in line with a meta-analysis by Bai et al., who showed an age of over 65 years as a predictor for post-TIPS HE, which was also shown in a single-center study by Li et al. [15,16]. Further analyses of potential predictors of a post-TIPS HE revealed predominantly heterogeneous results [15,26,27,28,29].

In cases where the post-TIPS HE cannot successfully be treated by diet or medication, patients with refractory HE suffered from increasing deterioration of liver function and the associated poor prognosis [14,17]. Therefore, we subsequently studied the efficacy of TIPS modification in the management of refractory post-TIPS HE. TIPS modification is increasingly integrated in multimodal treatment settings to avoid or delay liver transplantation [13,18,19]. Within these concepts it is important to be aware of patient safety in the course of TIPS modification. Although a variety of complications can occur after TIPS modification, particularly regarding the recurrence of the primary TIPS indication, especially variceal bleeding and ascites, there is limited evidence on the effectiveness and safety of TIPS modification [14,18,19].

In our study, we identified 5.8% of patients with refractory post-TIPS HE. This is in line with previous studies describing an incidence of refractory post-TIPS HE of up to 8% [13,14,18,19]. In the majority of patients in the cohort of Kochar et al., shunt modification was performed by a complete TIPS occlusion (*n* = 29/38, 76%), and only seven patients (24%) received a TIPS reduction. In contrast, in our study, where the majority (*n* = 18/20, 90%) received a reduction stent to reduce the lumen to a defined 5mm diameter, only *n* = 2 shunts (10%) were entirely occluded [18]. While shunt reduction at the time of the study by Kochar et al. was still a technical challenge, today, ready-to-use reduction stents offer the opportunity to easily downsize the shunts in a standardized manner. Within the patient cohort of Kochar et al. *n* = 3 deaths related to immediate procedure-associated complications occurred. Moreover, in this cohort, *n* = 8 patients died within one week after TIPS occlusion, most likely due to a sudden increase in portal venous pressure leading to venous bowel ischemia. In contrast, there were no procedure-related complications in our study. This suggests that reduction stents should be considered before an entire TIPS occlusion is performed. In our study, a response to shunt modification (defined as an improvement of HE of at least one grade according to the West Haven criteria) was found in 55% of cases and non-response in 45%, which is comparable to the results of Kochar et al. (58% responder, 42% non-responder). In our cohort, 91% of the responders survived the 6 months follow-up compared with only 67% of the non-responders (this difference not being statistically significant), while in the study of Kochar et al., 80% of the responders survived the 6 month follow-ups versus 27% of non-responders. Of note, although not statistically significant, the group of non-responders in our cohort showed decreased liver function and more advanced stages of cirrhosis. Kochar et al. could not identify an independent predictive variable for response to shunt modification in their study. In contrast, we were able to show that patients with severe HE (West Haven grade 3/4) benefit from shunt modification, identifying a higher HE grade after TIPS as the only positive predictor (*p* = 0.019), independent of liver function and PPG.

In another single-center study, De Keyzer et al. described 21 of 347 patients (6%) with the indication for TIPS modification in refractory post-TIPS HE [19]. They differentiated patients into survivors within a 1 month- (71%) and 6 months (52%) follow-up and non-survivors, irrespective of HE development. In contrast, our study, similar to Kochar et al., shows different survival rates depending on the resulting HE grade.

In a comparable study design, Rowley et al. were able to identify a shunt diameter higher than 8 mm, a positive HE history, and lowered serum albumin levels as predictive factors for the development of refractory post-TIPS HE [14]. In our subgroup analysis, patients with a decreased serum albumin, a sign for impaired liver function before TIPS, were less likely to receive a TIPS modification in the later course, possibly indicating that an interaction of HE grade and liver function identifies the “optimal” patient for TIPS modification. HE was found to be improved in *n* = 8/10 (80%) cases.

Regarding the safety of shunt modification, we could show that shunt reduction to a diameter of 5mm does not lead to relapse of variceal bleeding or refractory ascites in the majority of patients, demonstrating again the reduction is safe and should preferably be performed compared to complete shunt occlusion.

This study is limited by its retrospective study design, including a potential selection bias. Results need to be validated in a prospective study with a larger number of cases, especially with regard to the heterogeneous results of the analyzed patient cohort.

## 5. Conclusions

We were able to delineate an older patient age as a predictor for the development of post-TIPS HE and identify increased serum creatinine levels and INR right after TIPS as further potential predictors for the development of shunt-induced HE. Moreover, TIPS reduction can be considered to be a safe treatment option with particular benefit for patients with severe HE, regardless of PPG and liver function. Shunt reduction, interestingly, is not associated with a relapse of the initial TIPS indication, such as variceal bleeding or ascites, and should therefore be favored over complete shunt occlusion.

## Figures and Tables

**Table 1 jcm-09-00567-t001:** Patient characteristics.

		No HE	HE	
Parameter	Total No. (%)	No. (%)	No. (%)	*p*-value
Total	93 (100)	48 (51.6)	45 (48.4)	
Sex				0.624
male	65 (69.9)	33 (68.8)	32 (71.1)	
female	28 (30.1)	15 (31.3)	13 (28.9)	
Median age (range), years	59 (21–77)	54 (21–77)	56 (25–77)	0.984
Aetiology of liver disease				0.232
alcoholic	47 (50.5)	25 (52.1)	22 (48.9)	
PBC/autoimmune	11 (11.8)	6 (12.5)	5 (11.1)	
viral	7 (7.5)	5 (10.4)	2 (4.4)	
NASH	9 (9.7)	3 (6.3)	6 (13.3)	
other	19 (20.4)	9 (18.8)	10 (22.2)	
Child’s grade				0.559
A	17 (18.3)	9 (18.8)	8 (17.8)	
B	58 (62.4)	31 (64.6)	27 (60.0)	
C	18 (19.4)	8 (16.7)	10 (22.2)	
Post-TIPS HE grade (West Haven criteria)				E
0	48 (51.6)	48 (100)	0 (0)	
1	19 (20.4)	0 (0)	19 (42.2)	
2	12 (12.9)	0 (0)	12 (26.7)	
3	9 (9.7)	0 (0)	9 (20.0)	
4	5 (5.4)	0 (0)	5 (11.1)	
Indication for TIPS insertion				0.294
variceal bleeding	23 (24.7)	13 (27.1)	10 (22.2)	
refractory ascites	64 (68.8)	31 (64.6)	33 (73.3)	
other	6 (6.5)	4 (8.3)	2 (4.4)	

Abbreviations: HE, hepatic encephalopathy; NASH, nonalcoholic steatohepatitis; PBC, primary biliary cholangitis; TIPS, transjugular intrahepatic portosystemic shunt; E, not calculated.

**Table 2 jcm-09-00567-t002:** Laboratory data and pressure measurements regarding TIPS insertion.

	No HE (*n* = 48)	HE (*n* = 45)	
	mean (SD)	mean (SD)	*p*-value
Before TIPS insertion			
Bilirubin, mg/dl	2.1 (2.2)	2.1 (3.6)	0.664
Albumin, g/dl	3.0 (1.1)	2.6 (1.2)	0.525
Creatinine, mg/dl	1.2 (0.7)	1.5 (1.2)	0.125
INR	1.4 (0.3)	1.3 (0.3)	0.475
MELD-score	14.1 (5.5)	14.8 (6.7)	0.347
CLIF-c AD-score	57.5 (8.8)	56.7 (9.4)	0.839
PPG, mmHg	18.4 (5.8)	18.8 (5.8)	0.973
After TIPS insertion			
Bilirubin, mg/dl	2.3 (1.5)	2.6 (4.1)	0.225
Albumin, g/dl	3.1 (0.8)	2.9 (0.6)	0.304
Creatinine, mg/dl	1.1 (0.9)	1.3 (0.9)	0.610
INR	1.4 (0.3)	1.4 (0.3)	0.687
MELD-score	12.1 (6.7)	13.4 (7.0)	0.668
CLIF-c AD-score	48.4 (6.6)	51.2 (5.9)	0.441
PPG, mmHg	9.0 (3.3)	8.9 (3.4)	0.249
PPG reduction, mmHg	9.4 (4.5)	9.9 (4.8)	0.654

Abbreviations: CLIF-c AD, CLIF Consortium Acute Decompensation; HE, hepatic encephalopathy; INR, internationalized normalized ratio; MELD, Model of End Stage Liver Disease; PPG, portal pressure gradient; SD, standard deviation; TIPS, transjugular intrahepatic portosystemic shunt.

**Table 3 jcm-09-00567-t003:** Results of logistic regression analysis of independent predictors for.

post-TIPS HE			
Variable	Regression Coefficient	Standard Effort	p-Value
Age	0.120	0.044	0.007
Post-TIPS Creatinine	4.934	2.099	0.019
Post-TIPS INR	4.000	1.943	0.040

Abbreviations: HE, hepatic encephalopathy; INR, internationalized normalized ratio; TIPS, transjugular intrahepatic portosystemic shunt.

**Table 4 jcm-09-00567-t004:** Subgroup laboratory data and pressure measurements before and after TIPS insertion.

	post-TIPS HE	post-TIPS HE	
	no Shunt Modification	Shunt Modification	
	*n* = 25	*n* = 20	
	mean (SD)	mean (SD)	*p*-value
Before TIPS insertion			
Bilirubin, mg/dl	2.5 (4.5)	2.0 (1.2)	0.944
Albumin, g/dl	2.2 (1.2)	3.2 (0.9)	0.014
Creatinine, mg/dl	1.7 (1.5)	1.2 (0.6)	0.251
INR	1.3 (0.3)	1.4 (0.3)	0.929
MELD-score	14.3 (1.8)	13.5 (4.7)	0.802
PPG, mmHg	20.8 (5.3)	16.0 (5.5)	0.016

After TIPS insertion			
Bilirubin, mg/dl	2.6 (5.0)	2.6 (1.6)	0.985
Albumin, g/dl	3.0 (0.7)	3.1 (0.3)	0.878
Creatinine, mg/dl	1.6 (1.0)	0.9 (0.4)	0.112
INR	1.3 (0.2)	1.5 (0.4)	0.059
MELD-score	13.7 (7.7)	13.0 (5.8)	0.766
PPG, mmHg	9.9 (3.3)	7.4 (3.0)	0.027
PPG reduction, mmHg	10.9 (4.9)	8.6 (4.4)	0.297

Abbreviations: HE, hepatic encephalopathy; INR, internationalized normalized ratio; MELD, Model of End Stage Liver Disease; PPG, portal pressure gradient; SD, standard deviation; TIPS, transjugular intrahepatic portosystemic shunt.

**Table 5 jcm-09-00567-t005:** Comparison of responders to TIPS modification and non-responders.

	Responder	Non-Responder	
	*n* = 11	*n* = 9	
Sex (m:f)	9:2	3:6	
Median Age (range), y	64 (56–77)	65 (26-76)	
Aetiology (ALD/non-ALD)	5/6	4/5	
Child’s grade (A/B/C)	5/4/2	2/5/2	
post-TIPS HE			
West Haven criteria (1–2/3–4)	1/11	9/0	
early (< 6 months) / late (> 6 months)	8/3	4/5	
TIPS reduction / occlusion	11/0	7/2	
Recurrence of primary TIPS indication	*n* = 1	*n* = 2	
	mean (SD)	mean (SD)	*p*-value
Before TIPS modification			
Bilirubin, mg/dl	1.9 (1.3)	3.4 (3.2)	0.149
Albumin, g/dl	3.1 (0.5)	2.9 (0.5)	0.746
Creatinine, mg/dl	1.5 (0.7)	1.8 (1.9)	0.111
INR	1.4 (0.4)	1.6 (0.5)	0.839
MELD-score	15.4 (5.6)	18.3 (6.3)	0.928
CLIF-c AD-score	52.7 (5.6)	55.2 (8.7)	0.565
PPG, mmHg	7.3 (3.1)	8.2 (5.1)	0.182
After TIPS modification			
Bilirubin, mg/dl	1.9 (1.7)	2.0 (1.0)	0.215
Albumin, g/dl	3.3 (0.6)	3.1 (0.5)	0.559
Creatinine, mg/dl	1.6 (1.2)	1.5 (1.4)	0.714
INR	1.4 (0.4)	1.5 (0.3)	0.490
MELD-score	15.6 (6.5)	17.1 (6.7)	0.864
CLIF-c AD-score	53.0 (5.1)	54.3 (8.4)	0.299
PPG, mmHg	11.8 (4.2)	12.6 (4.9)	0.660
PPG increase, mmHg	3.8 (4.3)	5.6 (4.9)	0.862
XA time, min	4.0 (2.7)	6.8 (3.5)	0.457

Abbreviations: ALD, alcoholic liver disease; CLIF-c AD, CLIF Consortium Acute Decompensation; HE, hepatic encephalopathy; INR, internationalized normalized ratio; MELD, Model of End Stage Liver Disease; PPG, portal pressure gradient; SD, standard deviation; TIPS, transjugular intrahepatic portosystemic shunt; XA, fluoroscopy

**Table 6 jcm-09-00567-t006:** Results of logistic regression analysis of independent predictors for response of TIPS modification.

Variable	Regression Coefficient	Standard Effort	p-Value
HE grade (West Haven)	2.634	1.125	0.019

Abbreviations: HE, hepatic encephalopathy; TIPS, transjugular intrahepatic portosystemic shunt.

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
