# Peer review of "TIPS Modification in the Management of Shunt-Induced Hepatic Encephalopathy: Analysis of Predictive Factors and Outcome with Shunt Modification"

_jcm, 2020, doi:10.3390/jcm9020567_

Round 1
Reviewer 1 Report
The authors state that in this retrospective review they recorded grade of encephalopathy. First, it seems unusual grade of encephalopathy was routinely recorded in every chart- how many patients had missing data for grade of encephalopathy?
Was grade 1 encephalopathy recorded which would require psychometric testing? It seems more likely that either no encephalopathy or grade 2 or greater was recorded and that most patients with grade 3 or 4 encephalopathy prior to TIPS would not have proceeded with TIPS.
In Table 1 provide grade of encephalopathy for the HE group.
Table 4 provide mean lactulose dose and rifaximin dose for each group
Table 5- why would a patient need TIPS modification if they only had garde 1-2 HE post TIPS? Couldn’t this be managed medically?
Table 5- TIPS modification- do you mean downsizing of the TIPS?
What were the causes of death in the nonresponders?
T he authors state that cases were matched to controls by propensity score matching “TIPS were identified by case-control-matching (propensity score matching) 68 regarding age, sex, etiology of liver disease, child grade and indication for TIPS insertion. 69 Laboratory data (Bilirubin, Albumin, INR, Creatinine” If matching is performed then appropriate statistical tecnniques must be applied for regression. Please state the specific multivariable analysis performed.
Would add the following references that are relevant to this topic
1: Rowley MW, Choi M, Chen S, Hirsch K, Seetharam AB. Refractory Hepatic
Encephalopathy After Elective Transjugular Intrahepatic Portosystemic Shunt: Risk
Factors and Outcomes with Revision. Cardiovasc Intervent Radiol. 2018
Nov;41(11):1765-1772. doi: 10.1007/s00270-018-1992-2. Epub 2018 Jun 5. PubMed
PMID: 29872892.
2: Nardelli S, Gioia S, Pasquale C, Pentassuglio I, Farcomeni A, Merli M,
Salvatori FM, Nikolli L, Torrisi S, Greco F, Nicoletti V, Riggio O. Cognitive
Impairment Predicts The Occurrence Of Hepatic Encephalopathy After Transjugular
Intrahepatic Portosystemic Shunt. Am J Gastroenterol. 2016 Apr;111(4):523-8. doi:
10.1038/ajg.2016.29. Epub 2016 Mar 1. PubMed PMID: 26925879.
3: Merola J, Chaudhary N, Qian M, Jow A, Barboza K, Charles H, Teperman L, Sigal
Hyponatremia: A Risk Factor for Early Overt Encephalopathy after TransjugularIntrahepatic Portosystemic Shunt Creation. J Clin Med. 2014 Apr 4;3(2):359-72.
doi: 10.3390/jcm3020359. PubMed PMID: 26237379; PubMed Central PMCID: PMC4449686.
4: Russo MW, Sood A, Jacobson IM, Brown RS Jr. Transjugular intrahepatic
portosystemic shunt for refractory ascites: an analysis of the literature on
efficacy, morbidity, and mortality. Am J Gastroenterol. 2003 Nov;98(11):2521-7.
Review. PubMed PMID: 14638358.
Minor comments-
higher patient age should be replaced by older patient age
add survived to end of the sentence in abstract 10/11 responders (91%) survived the 6-month follow-up 28 after modification but only 6/9 non-responders (67%) survived.
Proofread and correct minor grammatical mistakes
Author Response
Dear Reviewers, honored members of the editorial board,
we would like to express our gratitude for the positive evaluation of our manuscript and especially for the thoughtful comments made. The points raised by the reviewers helped us to hopefully further improve our manuscript. Each point of the reviewers’ points was considered and we provide a detailed point-by-point reply below. The modifications in the manuscript a marked using the track-changes version. Again, we would like to thank the reviewers for their comments and hope that the manuscript can now be accepted by the Journal of Clinical Medicine.
Sincerely yours,
Moritz Wildgruber on behalf of the authors
Reviewer 1:
The authors state that in this retrospective review they recorded grade of encephalopathy. First, it seems unusual grade of encephalopathy was routinely recorded in every chart- how many patients had missing data for grade of encephalopathy?
RE: In fact, we had actually no missing data on HE levels. The continuous HE grading is part of an active surveillance program at our center. All patients with progressive cirrhosis are included in the evaluation for liver transplantation, thus close monitoring of these patients is being performed routinely, which in our case was helpful to gain data without missing values.
Was grade 1 encephalopathy recorded which would require psychometric testing? It seems more likely that either no encephalopathy or grade 2 or greater was recorded and that most patients with grade 3 or 4 encephalopathy prior to TIPS would not have proceeded with TIPS.
RE: Pre-TIPS, all patients routinely underwent a paper and pencil test using a number
connection test (psychometric testing). Minimal hepatic encephalopathy was assumed when
patients took longer than 30 seconds to correctly complete the test. Regarding the second point of the reviewer: ‘patients with grade 3 or 4 encephalopathy prior to TIPS would not have proceeded with TIPS’ - we absolutely agree, patient with manifest encephalopathy are not suited to undergo TIPS implantation, and in our institution this is regarded even a contra-indication.
In Table 1 provide grade of encephalopathy for the HE group.
RE: We added the grade of encephalopathy for the HE group as requested.
Table 4 provide mean lactulose dose and rifaximin dose for each group.
RE: As described in the methods section (2.1 Study Design), all patients who received shunt
modification had undergone maximal medical treatment previously, including lactulose and
rifaximin, according to current guidelines (Rifaximin 550 mg 1-0-1, Lactulose 3x 15-20 ml).
There were no different treatment strategies between the groups. We added the requested dose levels to the methods section.
Table 5- why would a patient need TIPS modification if they only had grade 1-2 HE post TIPS? Couldn’t this be managed medically?
RE: We agree with the reviewer that for grade 1-2 HE medical management is the strategy of choice. However, if HE persists under optimized medical treatment, then TIPS modification is indicated. This was the case for the patients in our cohort with lower HE grades. TIPS modification was only performed after failed medical management.
Table 5- TIPS modification- do you mean downsizing of the TIPS?
RE: TIPS modification in most cases means downsizing. However as in n=2 patients the TIPS was not downsized, but had to be completely occluded (see point 3.5 in the manuscript), we thought that the term ‘downsizing’ is not entirely correct, downsizing in n=2 cases would thus mean downsizing down to zero. We would suggest to stick to ’modification’ but we’re open if the reviewer thinks ‘downsizing’ is the better term.
What were the causes of death in the nonresponders?
RE: N=2 patients of the non-responders died of liver failure, n=1 patient died during liver transplant surgery after unexplained resuscitation status.
The authors state that cases were matched to controls by propensity score matching “TIPS were identified by case-control-matching (propensity score matching) 68 regarding age, sex, etiology of liver disease, child grade and indication for TIPS insertion. 69 Laboratory data (Bilirubin, Albumin, INR, Creatinine” If matching is performed then appropriate statistical techniques must be applied for regression. Please state the specific multivariable analysis performed.
RE: As requested, we describe the matching process in more detail, and added an extended paragraph to the methods section (2.4 Statistics).
Would add the following references that are relevant to this topic
1: Rowley MW, Choi M, Chen S, Hirsch K, Seetharam AB. Refractory Hepatic
Encephalopathy After Elective Transjugular Intrahepatic Portosystemic Shunt: Risk
Factors and Outcomes with Revision. Cardiovasc Intervent Radiol. 2018
Nov;41(11):1765-1772. doi: 10.1007/s00270-018-1992-2. Epub 2018 Jun 5. PubMed
PMID: 29872892.
RE: This reference is already part of the manuscript and is being discussed in the third paragraph of the discussion section (ref 14.)
2: Nardelli S, Gioia S, Pasquale C, Pentassuglio I, Farcomeni A, Merli M,
Salvatori FM, Nikolli L, Torrisi S, Greco F, Nicoletti V, Riggio O. Cognitive
Impairment Predicts The Occurrence Of Hepatic Encephalopathy After Transjugular
Intrahepatic Portosystemic Shunt. Am J Gastroenterol. 2016 Apr;111(4):523-8. doi:
10.1038/ajg.2016.29. Epub 2016 Mar 1. PubMed PMID: 26925879.
RE: As described, it is of great clinical importance to identify predictive factors for the development of HE after TIPS, which allows patient selection based on a specific risk profile as well as targeting potential risk factors right after TIPS insertion. Beside the results of our study, further analyses of potential predictors of a post-TIPS HE revealed that pre-TIPS minimal hepatic encephalopathy was predictive of post-TIPS overt HE. We are thus happy to add the requested reference to the discussion section to underline this point.
3: Merola J, Chaudhary N, Qian M, Jow A, Barboza K, Charles H, Teperman L, Sigal
Hyponatremia: A Risk Factor for Early Overt Encephalopathy after Transjugular
Intrahepatic Portosystemic Shunt Creation. J Clin Med. 2014 Apr 4;3(2):359-72.
doi: 10.3390/jcm3020359. PubMed PMID: 26237379; PubMed Central PMCID: PMC4449686.
RE: Merola et al. were able to delineate hyponatremia before TIPS as another predictive factor for the development of HE. As described, our study is limited by its retrospective design and we cannot map all of the predictive factors that have already been discussed. A systemic review of predictive factors for the development of HE by Bai et al. did not consider the sodium level, which again underlines the heterogeneity of the results, as described in the discussion section. As in hyponatremic patients TIPS should be undertaken with special caution, we added the requested reference to the discussion section.
4: Russo MW, Sood A, Jacobson IM, Brown RS Jr. Transjugular intrahepatic
portosystemic shunt for refractory ascites: an analysis of the literature on
efficacy, morbidity, and mortality. Am J Gastroenterol. 2003 Nov;98(11):2521-7.
Review. PubMed PMID: 14638358.
RE: The reference was added to the introduction section.
Minor comments:
higher patient age should be replaced by older patient age
RE: We revised the term appropriately.
add survived to end of the sentence in abstract 10/11 responders (91%) survived the 6-month follow-up 28 after modification but only 6/9 non-responders (67%) survived.
RE: We revised the wording accordingly.
Reviewer 2 Report
The authors performed a retrospective analysis of patients with TIPS (n=344). And the authors found a higher patient's age, increased serum creatinine and elevated INR immediately after TIPS placement were independent predictors for the development of HE.
Although it is a single center study, the experience of shunt modification to manage shunt-induced hepatic encephalopathy has clinical relevance.
Since the importance of the Acute-on-Chronic Liver Failure (ACLF) is increasing, additional analyses of ACLF Scores and Grades are worth doing.
Author Response
Dear Reviewers, honored members of the editorial board,
we would like to express our gratitude for the positive evaluation of our manuscript and especially for the thoughtful comments made. The points raised by the reviewers helped us to hopefully further improve our manuscript. Each point of the reviewers’ points was considered and we provide a detailed point-by-point reply below. The modifications in the manuscript a marked using the track-changes version. Again, we would like to thank the reviewers for their comments and hope that the manuscript can now be accepted by the Journal of Clinical Medicine.
Sincerely yours,
Moritz Wildgruber on behalf of the authors
Reviewer 2:
The authors performed a retrospective analysis of patients with TIPS (n=344). And the authors found a higher patient's age, increased serum creatinine and elevated INR immediately after TIPS placement were independent predictors for the development of HE.
Although it is a single center study, the experience of shunt modification to manage shunt-induced hepatic encephalopathy has clinical relevance.
Since the importance of the Acute-on-Chronic Liver Failure (ACLF) is increasing, additional analyses of ACLF Scores and Grades are worth doing.
RE: we agree with the reviewer on the rising importance of ACLF and that novel grading scales are worthwhile to estimate the prognosis of these patients. As in our cohort acute liver failure was not frequently observed, ACLF grades were low (some parameters feeding into the score were negative in most patients, such as renal replacement therapy or mechanical ventilation) and thus computing statistics with this score being in the lower range did not produce meaningful results. However, we took the advice of the reviewer and computed the corresponding CLIF-C AD score. This was not statistically different between the HE and the non HE group but nevertheless the new values are now included in Table 2.